# Dr. Tetsuro Fujiwara—My Memories from the Early Days of Dr. Fujiwara’s Research

**DOI:** 10.3390/biomedicines12010218

**Published:** 2024-01-18

**Authors:** Alan H. Jobe

**Affiliations:** Perinatal Institute, Cincinnati Children’s Hospital Medical Center, University of Cincinnati School of Medicine, Cincinnati, OH 45229, USA; alan.jobe@cchmc.org; Tel.: +1-513-702-3730

**Keywords:** Fujiwara, surfactant, mortality, prematurity

## Abstract

This brief commentary honors Dr. Tesuro Fujiwara, the first person to treat infants with respiratory distress syndrome by instilling surfactant into their trachea. In the 1960s, mortality from RDS, which could only be treated with oxygen, was about 50 percent. Based on the physiology Fujiwara learned that lung immaturity could be treated with doses of surfactant from animals in sheep models. He then made a surfactant from cow lungs called Servanta and treated 10 infants with RDS, who all had a large improvement in oxygenation. Other new therapies, such as continuous positive airway pressure and newer infant ventilators—in combination with surfactant therapy have decreased infant mortality to less than 1% from RDS in the most recent US infant death statistics.

In the 1960s, mortality from respiratory distress syndrome (RDS), even in large preterm infants >2 kg or >34 weeks gestation, was extremely high, with the only therapy being supplemental oxygen. Following Avery and Mead’s identification of surfactants as causality for RDS, there were no clinical trials of surfactants over that 12-year gap from 1968 to 1980; however, there were multiple animal studies to learn how to use surfactants in babies [1]. Dr. Forest Adams was a well-known pediatric cardiologist then, and I asked why he was involved with surfactants. At that time, people thought that respiratory distress syndrome was a syndrome of pulmonary hypoperfusion and, therefore, a cardiac abnormality. Dr. Tetsuro Fujiwara was a visiting scientist with Dr. Adams at the University of California, Los Angeles (UCLA). They had a postdoctoral researcher, Machiko Ikegami, working with them then [2]. Dr. Adams retired and moved to Hawaii, leaving Dr. Ikegami without supervision, so I took her on as a fellow at Harbor UCLA. Dr. Ikegami and I worked productively together for about 35 years [3].

Dr. Fujiwara had been treating preterm sheep with sheep surfactant at UCLA, showing the benefit of intratracheal surfactant [2]. With Dr. Ikegami’s help, I started applying these techniques at Harbor UCLA. The surfactant was recovered by lavage from newborn lambs by a centrifugation procedure, which we then adapted to large-scale sheep studies using either surfactant from adult sheep or newborn lambs or newborn calves. The advantage of the sheep surfactant was that it was readily available, and we also used surfactant from calf lung [4].

Dr. Fujiwara returned to Japan and wanted to help babies, but he had no surfactant. He tried to obtain surfactant from cow lungs, but the law in Japan was that one had to slice lungs to ensure no tuberculosis was present. He could not lavage the lungs to retrieve surfactant when he did that. So, he developed a surfactant as a lipid extract of cow lung using an organic solvent extraction procedure and supplemented it with palmitic acid to improve surface ternion properties. Dr. Fujiwara developed extracted surfactant as an extract with chloroform–methanol that he could reconstitute in saline using surfactant from whole cow lungs. This surfactant ultimately became Survanta^®^. This surfactant was tested on rabbit models, while at the same time, Dr. Bengt Robertson in Stockholm and Dr. Goran Enhorning in Toronto were isolating surfactants as lipid extracts and treating animal models [5,6]. Dr. Robertson developed techniques for ventilating preterm rabbits, a real technological breakthrough in advancing the ability to make physiologic measurements. I went to Stockholm, and he taught me the techniques to do that. They built their own ventilators and published papers on ventilated preterm rabbits and how surfactants worked in the preterm lungs [7,8,9]. Another person who became interested in this field was Dr. Burkhard Lachmann, an East German scientist who defected to the West and began working at Rotterdam University [10]. He was also developing an extract surfactant from the lungs of animals. Dr. Lachmann organized a meeting on surfactants every few years. During the first one, we met in his laboratory at Rotterdam University, and he was trying to test his surfactant against Dr. Fujiwara’s surfactant. I recall that Dr. Lachmann used a strategy of high pressure to open the open, and when he did that, his surfactant worked reasonably well, but Dr. Fujiwara’s surfactant worked much better from low pressure. Therefore, he could open the lung with low pressure and without injuring the lung. They had a heated discussion about this, but Dr. Fujiwara was correct in gently ventilating and minimizing the injury to the preterm lung; this was an exciting encounter in the 1970s, but I cannot remember the exact date of that meeting.

Dr. Fujiwara returned to Japan, worked more on his surfactant, and was the first to treat babies with RDS; this discovery was published in the Lancet, and got everybody excited [11]. Robertson and his team kept working on better ways to manufacture surfactants. The Swedish surfactant developed in the Karolinska Institute by Tore Curstedt and Bengt Robertson became Curosurf^®^, and was used to treat babies all over Europe; this was very effective. Ultimately, it was tested against Survanta^®^, and was shown to be very similar [12].

Survanta^®^ proved to be a very effective surfactant made from cow lungs. Unfortunately, bovine spongiform encephalopathy disease (BSE) became an issue. This Prion disease could be avoided if cow lungs were from countries without BSE [13]. That became a problem in Europe since, there was no way to prove that the surfactant did not contain the Prion from cow lungs. Survanta^®,^ manufactured in the US came from calf lungs from New Zealand, so there were no concerns regarding BSE. Therefore, Dr. Fujiwara was the first to prove that intra-tracheal installation of surfactant was possible, and it would work. Surfactant treatments for RDS are now the standard of care worldwide. This surfactant was the first drug ever given to babies by intra-tracheal installation. This was a significant issue for drug regulators, because it represented a new biological route from animal sources into preterm infants. Survanta is still widely used worldwide, and Dr. Fujiwara should be recognized as a pioneer for his noteworthy investigations and contributions to the care of preterm infants with RDS.

In conclusion, the following four interventions have made respiratory distress syndrome no longer lethal in reasonable-sized infants:Surfactant [11];Continuous positive airway pressure [14];Antenatal steroids to decrease the incidence of RDS [15];Improved technology to ventilate infants [16].

Thank you, Dr. Fujiwara.

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
