# Peer review of "Dr. Tetsuro Fujiwara—My Memories from the Early Days of Dr. Fujiwara’s Research"

_biomedicines, 2024, doi:10.3390/biomedicines12010218_

Round 1

Reviewer 1 Report

Comments and Suggestions for Authors Thank you for asking me to review Dr Alan H Jobe' brief review on Prof Fujiwara's accomplishments. I red this commentary: I accept it without corrections It briefly tells, "how it all happens" with a personal flavor of Alan Jobe.

Author Response

No changes or comments are needed.

Reviewer 2 Report

Comments and Suggestions for Authors

The commentary by Jobe nicely relates the historical state of knowledge regarding surfactant, key investigators contributing to research advances toward clinical application, and the seminal work of Fujiwara.  I have a few suggestions:

1.     The manuscript needs editing to improve sentence structure and flow of content in several areas.

2.     Spelling error in the title: Fujiwara’s, and I suggest removing the ().

3.     It would be useful to the reader to include some more dates of key events to illustrate the historical perspective.

4.     Consider adding a comment about work of Avery and Mead (1959) first showing surfactant deficiency in hyaline membrane disease, which will illustrate the time lag before the first successful clinical application of replacement surfactant (1980).

5.     It might be appropriate to mention the importance of using natural surfactant, which contains the surfactant-associated proteins, versus lipids alone that were not effective in the first clinical study (Chu 1967).

6.     I agree with the conclusion regarding 4 advances that have markedly improved outcome for premature infants, and it’s interesting that they all occurred during the same era.

Comments on the Quality of English Language

1.     The manuscript needs editing to improve sentence structure and flow of content in several areas.

2.     Spelling error in the title: Fujiwara’s, and I suggest removing the ().

Author Response

I have added a reference to Avery and Mead. It's a good suggestion; there are 12 years between that discovery and an attempt at therapy.  This commentary was never meant to be a complete history.  Animal work was being done during that 12-year lag. The reviewer mentions a Chu reference from 1967.  I am unfamiliar with that reference; we searched PubMed for Chu – 1967 and other years and found nothing related to surfactants. So, no changes were made regarding that comment, which is highlighted in the following sentences about the sheep research conducted by Fujiwara at UCLA. The paper has been carefully reviewed for readability and spelling.

Reviewer 3 Report

Comments and Suggestions for Authors

Thank you for contributing to this special edition. There is some interesting material to share in this piece of writing, for instance I had not been aware of the issue of BSE in cow lungs affecting their use for surfactant, but it needs to be re-written for clarity. 

The first paragraph does not make sense and needs to be re-written. It would be good to start with a description of RDS and the associated morbidity and mortality prior to the development of surfactant. 

There is some repetition notably the reference to the key article published in the Lancet, which is discussed both in paragraph 3 and 5. The timeline needs to be revised, as the paper talks about work in the 70s, then the paper in the 80s before going back to the 60s. 

First line paragraph 2 repeats the word sheep. 

The conclusion is currently a list of bullet points, which do not relate to the material in the commentary, please re-write as a paragraph that summarises the material in the paper, and talks about the future of surfactant therapy or Dr Fujiwara's contribution. 

Comments on the Quality of English Language

The writing has some grammatical errors. 

Author Response

As requested, I have added a sentence about the high mortality from RDS before the 1960s. The first paragraph has more of a timeline and is more informative. Of the two sheep,  one is sheep and is correct, and the second one should proceed with, as with animals, they were treated with sheep surfactant.

I like the bullet points at the end and choose not to convert them to whole paragraphs, which would expand the paper considerably; based on the conclusion, perhaps numbering might be more appropriate. I have intentionally not introduced any discussion of the surfactant proteins, as that is a giant separate issue not associated with Fujiwara research.

Reviewer 4 Report

Comments and Suggestions for Authors

This is a lovely memoir from Professor Alan Jobe, a fitting testimony to Dr Fujiwara, who was a pioneer in surfactant research.

The commentary may well have been dictated, then typed by someone else... as it comes across as a verbal story, and has some repetition, and some spelling errors.  To make things easier I have cut and pasted and tried to remedy the flow of information to make it more readable, whilst at the same time being aware of the fact that this is a personal memoir from another great man in the field.

Comments on the Quality of English Language

I have edited minor typos and changed sequence of one reference

Author Response

Thank you for your kind comments and effort to improve the paper; we have incorporated them in the revision.

Round 2

Reviewer 3 Report

Comments and Suggestions for Authors

Thank you for revising this commentary which provides an interesting history on the development of surfactant. 

I have a few suggestions:

1. The second sentence is incomplete - "Following Avery and Mead's ...." What happened following their discovery?

2. In paragraph 3, it says at the beginning that whole cow lung lavage could not be done in Japan due to the TB laws. But then in line 33 it says that he used extract from whole cow lungs - did they change the law in Japan?

3. Please provide a reference for the statement that Curosurf and Survanta are of similar effectiveness, line 58. 

4. Line 62 - ref 12 is to prion disease in deer, please provide a reference that is relevant to cows. 

5. I would still prefer the Conclusion to be written as prose rather than as a numbered list. 

Comments on the Quality of English Language

The English is of good quality. No full stop needed after kg, line 8. Sentence beginning in line 9 needs to be completed. 

Author Response

  1. I have completed the sentence.
  2. He did not lavage the cow lungs; he used extracts from whole cow lungs.
  3. Reference provided (PMID 8808165)
  4. Provided new reference (PMID 26354437)
  5. I would prefer to keep it as is. 

Thank you - Alan H. Jobe, MD, PhD
